# NanoFIRE: A NanoLuciferase and Fluorescent Integrated Reporter Element for Robust and Sensitive Investigation of HIF and Other Signalling Pathways

**DOI:** 10.3390/biom13101545

**Published:** 2023-10-19

**Authors:** Alison E. Roennfeldt, Timothy P. Allen, Brooke N. Trowbridge, Michael R. Beard, Murray L. Whitelaw, Darryl L. Russell, David C. Bersten, Daniel J. Peet

**Affiliations:** 1School of Biological Sciences, The University of Adelaide, Adelaide, SA 5005, Australia; alison.roennfeldt@adelaide.edu.au (A.E.R.); timothy.allen@adelaide.edu.au (T.P.A.); brooke.trowbridge@adelaide.edu.au (B.N.T.); michael.beard@adelaide.edu.au (M.R.B.); murray.whitelaw@adelaide.edu.au (M.L.W.); 2Robinson Research Institute, School of Biomedicine, The University of Adelaide, Adelaide, SA 5006, Australia; darryl.russell@adelaide.edu.au; 3Research Centre for Infectious Diseases, School of Biological Sciences, The University of Adelaide, Adelaide, SA 5005, Australia; 4ASEAN Microbiome Nutrition Centre, National Neuroscience Institute, Singapore 169857, Singapore

**Keywords:** hypoxia, HIF, reporter system, cell-based, NanoLuciferase, lentiviral, transcription, high throughput assay

## Abstract

The Hypoxia Inducible Factor (HIF) transcription factors are imperative for cell adaption to low oxygen conditions and development; however, they also contribute to ischaemic disease and cancer. To identify novel genetic regulators which target the HIF pathway or small molecules for therapeutic use, cell-based reporter systems are commonly used. Here, we present a new, highly sensitive and versatile reporter system, NanoFIRE: a NanoLuciferase and Fluorescent Integrated Reporter Element. Under the control of a Hypoxic Response Element (HRE-NanoFIRE), this system is a robust sensor of HIF activity within cells and potently responds to both hypoxia and chemical inducers of the HIF pathway in a highly reproducible and sensitive manner, consistently achieving 20 to 150-fold induction across different cell types and a Z′ score > 0.5. We demonstrate that the NanoFIRE system is adaptable via substitution of the response element controlling NanoLuciferase and show that it can report on the activity of the transcriptional regulator Factor Inhibiting HIF, and an unrelated transcription factor, the Progesterone Receptor. Furthermore, the lentivirus-mediated stable integration of NanoFIRE highlights the versatility of this system across a wide range of cell types, including primary cells. Together, these findings demonstrate that NanoFIRE is a robust reporter system for the investigation of HIF and other transcription factor-mediated signalling pathways in cells, with applications in high throughput screening for the identification of novel small molecule and genetic regulators.

## 1. Introduction

The Hypoxia Inducible Factors 1 and 2 (HIF-1/2) are key transcription factors induced under low oxygen conditions. In normal physiology they are essential for processes including vascular development, erythropoiesis, and lung function [1,2,3]. However, the HIFs are also implicated in the pathophysiology of numerous diseases. In ischaemic diseases such as stroke and wound healing, the HIFs are advantageous and promote blood vessel growth for oxygen delivery, while in diseases such as chronic kidney disease, elevated HIF activity increases erythropoietin production to overcome associated anaemia [4]. However, the HIFs are also commonly pro-tumorigenic, driving angiogenesis and metabolic transformation, and are therefore attractive therapeutic targets for inhibition [5,6]. Thus, the HIFs are promising targets for both activation and inhibition, dependent on the disease context.

HIF regulation is predominately controlled post-translationally in an oxygen-dependent manner. HIF-1 and HIF-2 each consist of an oxygen-regulated HIF-α subunit (HIF-1α or HIF-2α, respectively), which under normoxic conditions is hydroxylated by Prolyl Hydroxylase Domain enzymes (PHDs) in an oxygen-dependent manner, initiating ubiquitination by the E3 ligase Von Hippel–Lindau (VHL) and resulting in proteasome-mediated degradation [7]. Factor inhibiting HIF (FIH) under normoxic conditions also hydroxylates the HIF-α subunits in an oxygen-dependent manner, which inhibits histone acetyltransferases p300 and CREB-binding protein (CBP) association, downregulating HIF transcriptional activity [8]. When oxygen levels decrease the oxygen-dependent PHDs and FIH are inhibited, such that the HIF-α proteins are stabilised and can bind coactivators p300 and CBP. HIF-α dimerises with its constitutively expressed partner protein ARNT (also known as HIF-β), forming the active HIF transcription factor, and binds to Hypoxic Response Elements (HREs) to drive gene transcription.

HIF-driven cell-based reporter systems have been used extensively to investigate the mechanisms of HIF regulation and to identify small molecules which target the pathway, with the goal of using this knowledge for therapeutic targeting of HIF. Transiently transfected firefly luciferase reporters are most commonly used [9,10,11]. However, despite displaying high sensitivity, their transient nature results in labour intensive setup for high throughput screens, they are restricted to cells that are efficiently transfected, and plasmid-based reporters lack the chromatin context of endogenous HIF target genes [12]. More recently, fluorescence-based HIF reporter systems which can be stably integrated into cells have been developed, acting as a more suitable system for high throughput screening (HTS) purposes [13,14,15,16]. However, these systems commonly lack the sensitivity of luminescence-based reporters, and most fluorophores require oxygen for optimal fluorescence, limiting their applications in the investigation of signalling under hypoxia [17].

To overcome these limitations, and thus develop a system which is stably integrated, is sensitive, and can be used under hypoxia, we developed NanoFIRE, a NanoLuciferase and Fluorescent Integrated Reporter Element, to act as a sensitive, stable reporter system to investigate HIF and other signalling pathways. By placing NanoFIRE under the control of a Hypoxic Response Element (HRE-NanoFIRE), we formed a robust sensor of HIF activity within cells. We demonstrate that HRE-NanoFIRE can respond to hypoxia and chemical inducers of the HIF pathway, dimethyloxalylglycine (DMOG) and FG-4592, within multiple cell lines and primary cells. NanoFIRE is also highly versatile, with adaption to a synthetic transcription factor system allowing investigation of transcriptional regulation by the HIF regulator FIH, and substitution of the upstream response element facilitating investigation of progesterone receptor signalling. We therefore present NanoFIRE as a sensitive and versatile reporter system which can be used to investigate transcription factor and transcriptional regulation activity across cell lines and primary cells.

## 2. Materials and Methods

### 2.1. Animals

21-day old CBA × C57BL/6 F1 (CBAF1) mice were obtained from the University of Adelaide Laboratory Animal Services. Mice were given water and chow ad libitum and maintained in 12 h light/12 h dark conditions. All experiments were approved by The University of Adelaide Animal Ethics Committee and were conducted in accordance with the Australian Code of Practice for the Care and Use of Animals for Scientific Purposes; ethics number M-2021-058.

### 2.2. Mammalian Cell Culture—Cell Lines

HEK-293T and U2OS cells were grown in Dulbecco’s modified Eagle Medium (DMEM) + pH 7.5 HEPES with 10% Foetal Bovine Serum (FBS; Corning, Silverwater, NSW, Australia), 1% penicillin/streptomycin (Thermo Fisher, Scoresby, VIC, Australia) and 1× GlutaMAX (Gibco, Sydney, NSW, Australia). The same medium was used to grow Huh7 cells but without GlutaMAX. KGN cells were grown in a 1:1 mix of DMEM pH 7.5 HEPES (Gibco) and F12 nutrient mix (Gibco) with 10% FBS, 1% penicillin/streptomycin and 1× GlutaMAX. All KGN cells used stably expressed the pLV-TET2PURO-PR(B) construct unless otherwise stated. All cells were maintained at 37 °C and 5% CO_2_ in a humidified incubator.

### 2.3. Generation of NanoFIRE and Other Expression Constructs

The NanoFIRE Lentiviral plasmid (NoRE) was generated by digesting pLV-REPORT(PGK) (Addgene #172328) [13] with *SalI*/*SpeI* removing the TomatoHSVTk-2A-Neo cassette and assembled using a G-block containing *NanoLuciferase* using Gibson assembly. The NanoFIRE plasmid was sequenced confirmed by whole plasmid sequencing (plasmidsaurus) and deposited to Addgene (Addgene #208576). To create HRE-NanoFIRE, the HRE (originally from Razorenova et al. [18], as described previously [13]) was excised from the dual fluorescent reporter plasmid pLV-REPORT(PGK/CMV)-12xHRE (Addgene #172335) using *ClaI*/*AscI* and ligated into *ClaI*/*AscI* digested NoRE-NanoFIRE. GalRE-NanoFIRE and PRE-NanoFIRE were likewise cloned using *ClaI*/*AscI* with the GalRE response element sequence obtained from the GalRE dual fluorescent reporter plasmid (Addgene #172336) and the PRE sequence from the PRE-dual fluorescent reporter plasmid [13]. The dox inducible gal4DBD-HIFCAD expression construct (pLV-TET2Puro-Gal4DBD-HIFCAD) was cloned as per Allen et al. [13] (Addgene #207173). PR(A) and PR(B) expression constructs were codon optimised with N-terminal 2xStrep-TEV-3xFlag tags, synthesised and cloned into pENTR(twist) by TwistBiosciences (South San Francisco, CA, USA). The PR(A) or PR(B) constructs were recombined into pLV-TET2Puro [13] using LR Clonase II (Thermo Fisher).

### 2.4. Lentivirus Production and Generation of Stable Cell Lines

First, 80% confluent HEK-293T cells were transfected with 8.2 μg psPAX (Addgene #12260), 3.75 μg pMD2G (Addgene #12259) and 12.5 μg of the required lentiviral construct using polyethyleneimine (PEI) at a 3 μg: 1 μg ratio with DNA in serum free DMEM or Optimem. Then, 16 h later, the media was changed, and virus harvested 1–2 days later and filtered (0.45 μM filter, Satorius, Göttingen, Germany). Target cells were transduced at a MOI < 1 and incubated with virus for 48 h. Media was then changed to fresh culture media containing required antibiotics for selection; hygromycin at 140 μg/mL (Thermo Fisher) or puromycin at 1 μg/mL (Sigma-Aldrich, Macquarie Park, NSW, Australia) and maintained under antibiotic selection for approximately 2 weeks. Lentivirus for the transduction of primary mouse granulosa cells was generated and concentrated by The University of Adelaide Gene Silencing & Expression facility (GSEx). Briefly, a similar protocol to the above was completed for lentivirus production but using HEK-293T/17 cells and Lipofectamine 3000 with Optimem as per the manufacturers protocol (Thermo Fisher). Virus was collected 24 and 48 h post media change and concentrated by ultracentrifugation. Virus was titred using HEK-293T/17 cells infected with serial dilutions of virus and assessed for EGFP expression using flow cytometry.

### 2.5. NanoLuciferase Reporter Assays—Stable Cell Lines

U2OS and HEK-293T cells were seeded at 0.5 × 10^4^ cells/well and KGN and Huh7 cells at 1 × 10^4^ cells/well in white walled, clear bottom 96 well plates, with all HEK-293T cells spun down and resuspended in fresh media prior to plating. The following day, cells were treated with 1 mM DMOG (Cayman Chemical, Ellsworth Ann Arbor, MI, USA) or 0.1% DMSO, 1 μg/mL doxycycline (Sigma-Aldrich, Australia) or 0.1% H_2_O, 50 μM FG-4592 (Cayman Chemical, USA) or 0.1% DMSO, and 100 nM R5020 (PerkinElmer, North Ryde, NSW, Australia) or 0.1% ethanol. Huh7 cells were treated with 0.1 mM DMOG due to 1 mM DMOG showing significant toxicity. Then, 16 h post treatment, reporter output was determined using the Nano-Glo Luciferase Assay System as per the manufacturer’s instructions with minor modifications (Promega, Alexandria, NSW, Australia). Briefly, cells were removed from the incubator for 10 min to allow room temperature equilibration, media aspirated, then 25 μL of assay buffer mixed 50:1 with NanoGlo substrate was injected into each well. After 3 min incubation, luminescence was recorded using the GloMax Discovery Microplate Reader (Promega). Each experiment was completed three times independently each in duplicate or triplicate. Where stated, NanoLuciferase units (NLU) were normalised to DMOG treated cells under normoxia and stated as normalised NLU (NNLU). Hypoxic cell incubations were completed in the Oxford Optronix HypoxyLab at stated oxygen concentrations under humidified conditions at 37 °C and 5% CO_2_.

### 2.6. Primary Granulosa Cell Culture and NanoLuciferase Reporter Assays

Primary mouse granulosa cells were isolated and cultured as described in Dinh et al. [19], but cultured in a 1:1 mix of F12:DMEM (no glucose) with 35 nM testosterone (Sigma Aldrich), 1 μM retinoic acid (Sigma-Aldrich), and 50 ng/μL recombinant mouse follicle stimulating hormone (R&D Systems, Minneapolis, MN, USA) and seeded onto fibronectin-coated, white-walled, clear-bottom 96 well plates at 70,000 cells/well. Cells were transduced with approximately 1.8 × 10^6^ infectious units of HRE-NanoFIRE or NoRE-NanoFIRE lentivirus in a total volume of 1 μL/well, incubated for 24 h, then treated with 1 mM DMOG (Cayman Chemical, USA) or 0.1% DMSO. Finally, 24 h post treatment, the NanoLuciferase reporter assay was completed as per above, but with 50 μL of NanoLuciferase reagent added per well.

### 2.7. High Content Imaging of Fluorescent Reporter Cells

Stable HEK-293T HIF dual fluorescent monoclonal reporter cells [13] were seeded in black, clear-bottom, 96 well plates at 1 × 10^4^ cells/well in cell culture media (DMEM with 10% FBS, 1× glutaMAX and 1% penicillin/streptomycin). Then, 24 h later, cells were treated with 1 mM DMOG, 50 μM FG-4592 or vehicle (0.1% DMSO). Next, 16 h post treatment, cell populations were imaged in media at 10× magnification using the Thermo Fisher ArrayScan^TM^ XTI High Content Reader. Tomato mean fluorescent intensity (MFI) and EGFP MFI were imaged with an excitation source of 560/25 nm and 485/20 nm, respectively. Individual cells were defined by nuclear EGFP expression, determined by isodata thresholding to filter out background objects and abnormal nuclei were also excluded. MFI is the average from 2000 individual nuclei per well. EGFP MFI was used to confirm no change between treatment groups and control. Image quantification was completed using Thermo Fisher HCS Studio^TM^ 3.0 Cell Analysis Software.

### 2.8. Statistical Analysis and Figures

All data are expressed as mean ± standard deviation and *p* values calculated by a one-way ANOVA with Tukey’s multiple comparisons or for pairwise analysis, an unpaired T test, and were calculated using GraphPad Prism v9. Schematic figures were made using biorender.com (accessed on 27 September 2023). Z′ calculations were made using the following formula, as described in Zhang et al. [20].
$$Z^{\prime }=1-\frac{\left(3\sigma_{C+}+3\sigma_{C-}\right)}{\left(\mu_{C+}-\mu_{C-}\right)}$$

## 3. Results

### 3.1. Design and Characterisation of HRE-NanoFIRE to Investigate HIF Signalling

We aimed to develop a sensitive, versatile HIF reporter system which would respond to hypoxia and allow stable integration into the genome of any cultured cells of interest. To achieve this, we modified our previously established lentiviral dual fluorescent reporter construct [13], such that the oxygen-sensitive *tomato* fluorescent reporter gene was replaced with the bright and oxygen insensitive bioluminescent *NanoLuciferase* [21], forming the NanoFIRE reporter system. Within NanoFIRE, *NanoLuciferase* is expressed in a signal-dependent manner while a downstream, independent constitutive promoter controls expression of a *hygromycin* resistance gene and enhanced green fluorescent protein (EGFP), acting as a dual selectable marker (Figure 1A,B). PEST-tagged NanoLuciferase was chosen for better temporal response to changes in transcriptional activity [22], while lentiviral delivery enabled stable integration into the genome to eliminate the need for transient transfection and to facilitate adaptation to multiple cell types.

To investigate endogenous HIF signalling, we inserted a HRE concatemer from Razorenova et al. [18] upstream of the *NanoLuciferase* reporter gene, forming HRE-NanoFIRE (Figure 1A,B). For initial characterisation of HRE-NanoFIRE, we selected the osteosarcoma derived U2OS cell line as U2OS cells display strong induction of HIF-1α protein in response to hypoxia [23,24] and robustly activate HRE controlled reporter systems [25].

### 3.2. HRE-NanoFIRE Displays Robust Reporter Response to Hypoxia and Hypoxia Mimetics

To initially characterise HRE-NanoFIRE, we assessed reporter response in cells treated with chemical inducers of the HIF pathway, testing both a pan 2-oxoglutarate dependent dioxygenase inhibitor DMOG and PHD-specific inhibitor FG-4592. In stably integrated polyclonal U2OS HRE-NanoFIRE cells, robust and consistent reporter activation to DMOG (159-fold, Z′ = 0.90 ± 0.06) and FG-4592 (118-fold, Z′ = 0.83 ± 0.09) was achieved (Figure 1C), thus demonstrating high system inducibility, with the response to FG-4592 confirming PHD-specific reporter control [26]. The high Z′ scores for both DMOG and FG-4592 confirm high system robustness and suitability for HTS [20]. DMOG and FG-4592 were unable to induce reporter activity in a NanoFIRE reporter construct which lacked a response element (NoRE-NanoFIRE), confirming HIF-specific activation of the HRE-NanoFIRE system (Figure 1C). Reporter induction was time-dependent with peak reporter activity observed within 12 h of DMOG treatment, which slowly declined over 48 h (Figure 1D). This decline reflects the short half-life of PEST tagged NanoLuciferase and highlights the ability to detect changes in transcriptional activity within a short period of time, in contrast with fluorescent proteins which typically require longer periods of expression for sufficient signal accumulation [21,27].

Importantly, hypoxia robustly induced a U2OS HRE-NanoFIRE response, increasing with the severity of hypoxia; 1% O_2_ induced a 58-fold increase (Z′ = 0.77 ± 0.11) and 0.1% O_2_ induced a 132-fold increase (Z′ = 0.81 ± 0.16) relative to vehicle-treated cells at normoxia (Figure 1E,F). This was achieved without a specific reoxygenation step (approximate 10-min normoxia exposure to allow for temperature equilibration only), in contrast to the typical ≥4 h required for recovery in fluorescent-based reporter systems [28]. The robust response of HRE-NanoFIRE to chemical inducers of HIF-1α and hypoxia highlights its suitability for HTS applications to identify HIF modulators and for investigating HIF signalling under moderate and severe hypoxia.

### 3.3. HRE-NanoFIRE Is an Effective HIF Reporter System in Diverse Cell Lines and Primary Cells

We investigated the versatility of HRE-NanoFIRE within multiple cellular contexts as HIF-α regulation in response to hypoxia and hypoxia mimetics can differ between cell lines of diverse lineages [9,29]. Firstly, we generated a HEK-293T HRE-NanoFIRE polyclonal cell line, acting as a HIF-1 specific reporter system given that HEK-293T cells do not express HIF-2α [9]. Similar to U2OS cells, HEK-293T HRE-NanoFIRE cells induced robust reporter activity in response to DMOG (37-fold increase, Z′ = 0.77 ± 0.08) and FG-4592 (27-fold increase, Z′ = 0.51 ± 0.45) (Figure 2A). HEK-293T HRE-NanoFIRE cells also displayed consistent induction by 1% O_2_ (9-fold increase, Z′ = 0.61 ± 0.17), although less than with DMOG in normoxia, confirming successful reporter induction in response to hypoxic stimuli (Figure 2B).

Next, we assessed HRE-NanoFIRE in the ovarian tumour cell line KGN and hepatoma cell line Huh7 to investigate system adaptability in other cell contexts. Both KGN HRE-NanoFIRE and Huh7 HRE-NanoFIRE cells exhibited robust responses to DMOG, FG-4592, and 1% O_2_ with consistent Z′ values greater than 0.5, demonstrating assay robustness and suitability for HTS (Figure 2D–G). It is worth noting that while all cells induced HRE-NanoFIRE activity with DMOG, FG-4592 and 1% O_2_, the relative level of induction over background varied between cells and treatments. For example, in comparison to all other cell lines tested, KGN HRE-NanoFIRE cells displayed lower fold induction in response to the PHD inhibitor FG-4592 and 1% O_2_ compared to DMOG. This is likely reflective of differences between cell types in the levels of HIF-1α, HIF-2α, PHD1-3, and FIH, and is consistent with other studies, showing that the HIFs are regulated in a cell-type-dependent manner [9,29,30]. 

Together with demonstrating versatility across cell lines, it was important to test HRE-NanoFIRE activity within primary cells, given that there are limited systems available which allow investigation of HIF transcriptional activity in a primary cell context, and these cells are typically difficult to transfect. Hence, HRE-NanoFIRE was tested in primary mouse granulosa cells of the ovary. Granulosa cells were collected from pregnant mare serum gonadotropin (PMSG)-stimulated mice and cultured in vitro (Figure 3A). When these primary granulosa cells were transduced with HRE-NanoFIRE virus and the next day treated with DMOG for 24 h, 41-fold reporter induction (Z′ = 0.62 ± 0.08) was achieved compared to vehicle treated cells (Figure 3B). Mouse granulosa cells transduced with the NoRE-NanoFIRE control displayed no reporter induction in response to DMOG, confirming HIF specific activation of HRE-NanoFIRE. 

### 3.4. HRE-NanoFIRE Is More Sensitive Than Equivalent Fluorescent Reporter Systems

Fluorescent-based reporter systems are a common alternative for investigating HIF transcriptional activity [14,15,16]. We therefore compared HEK-293T HRE-NanoFIRE cells to our similar HIF dual fluorescence reporter system (comparing Figure 2A to Figure 2C) [13]. Our HEK-293T HIF dual fluorescent cells express a stable reporter construct similar to HRE-NanoFIRE but with nuclear *tomato* in place of the *NanoLuciferase* reporter gene [13]. While both systems efficiently report on HIF transcriptional activity when incubated with hypoxia mimetics for 16 h, the NanoLuciferase expressing HEK-293T HRE-NanoFIRE cells induced much higher reporter output than the dual fluorescent HEK-293T cells (37-fold compared to 4-fold with DMOG and 27-fold compared to 2-fold with FG-4592, respectively) (Figure 2A,C). This confirmed the enhanced sensitivity of HRE-NanoFIRE as a HIF reporter system over equivalent fluorescent-based systems.

### 3.5. NanoFIRE Can Be Adapted to Investigate Transcriptional Regulators and Synthetic Transcription Factors

The HRE-NanoFIRE system provides a read out of total HIF transcriptional activity, with PHD-specific inhibitor FG-4592 treatment allowing the contribution of PHD-dependent regulation to be determined. There are a lack of cell-based systems available, however, which allow for the specific determination of the FIH contribution to HIF transcriptional control, particularly in a high-throughput setting. We therefore aimed to develop a synthetic transcription factor-controlled NanoFIRE system, based on previously established transient FIH reporter systems [8], to provide a read out of FIH-dependent regulation in a stable cell-based setting.

A commonly used 5× Gal4 response element (GalRE) was inserted in place of the HRE in NanoFIRE, forming GalRE-NanoFIRE, which was transduced into HEK-293T cells. Stable GalRE-NanoFIRE expression was paired with stable doxycycline (dox) inducible expression [13,31] of the synthetic transcription factor gal4 DNA binding domain fused to the HIF1α-CAD (Gal4DBD-HIFCAD). This Gal4DBD-HIFCAD construct was adapted from a transient luciferase assay system to investigate FIH regulation of HIF [8]. Together, this formed the HEK-293T FIH-NanoFIRE reporter cell line. Thus, in the FIH-NanoFIRE system, FIH-mediated hydroxylation of the HIFCAD inhibits coactivator association, while non-hydroxylated HIFCAD acts as a strong transcriptional activator, meaning FIH controls reporter output (Figure 4A).

Analysis of HEK-293T FIH-NanoFIRE cells demonstrated minimal reporter activity with either DMOG or dox alone, whereas treatment with both dox and DMOG induced a 28-fold increase in FIH-NanoFIRE reporter activity above dox alone, demonstrating robust reporter activation upon FIH inhibition (Figure 4B). This produced a Z′ score of 0.92 ± 0.03 confirming high signal to background separation, assay consistency, and suitability for HTS applications. The GalRE-NanoFIRE reporter did not respond in cells which lacked the gal4DBD-HIFCAD expression construct, confirming specific reporter control by FIH (Figure 4B).

HEK-293T FIH-NanoFIRE cells also induced robust reporter activity in response to hypoxia with dox-treated cells incubated at 1% O_2_ inducing a 13-fold increase (Z′ = 0.76 ± 0.13) in reporter output relative to dox-treated cells at normoxia, and 0.1% O_2_ inducing a 12-fold increase (Z′ = 0.58 ± 0.44) (Figure 4C,D). Similar to the U2OS HRE-NanoFIRE line, HEK-293T FIH-NanoFIRE reporter induction in response to DMOG was time-dependent, with maximum reporter signal obtained between 12 and 24 h of treatment (Figure 4E). When treated with the PHD-specific inhibitor FG-4592, no significant increase in reporter activity was observed, confirming FIH-NanoFIRE is specifically controlled by FIH and not influenced by the PHDs (Figure 4F). Treatment with the FIH-specific inhibitor dimethyl N-oxalyl-D-phenylalanine (DM-NOFD) [32] at concentrations up to 1 mM produced no significant response (Figure 4F). This is consistent with our use of DM-NOFD in other cell-based assays where we have observed ineffective inhibition of FIH at concentrations up to 1 mM.

### 3.6. NanoFIRE Can Be Adapted to Investigate Other Transcription Factors

Finally, we aimed to investigate if the NanoFIRE system is more widely adaptable and could be used to investigate other transcription factors, specifically the Progesterone Receptor (PR). PR is a nuclear receptor consisting of isoforms PR(A) and PR(B). Upon cytoplasmic binding to its cognate ligand progesterone or the synthetic agonist R5020, PR undergoes nuclear translocation, homodimerisation, and DNA binding to cognate response elements to drive the expression of target genes [33]. PR is essential to female ovulatory control [34,35]; however, it is expressed in a limited number of cell lines making it difficult to investigate in this context [36,37]. We therefore paired expression of NanoFIRE under the control of a Progesterone Response Element (PRE-NanoFIRE) with stable expression of dox-inducible PR(A) or PR(B) in the ovarian granulosa cell cancer line KGN (Figure 5A), forming the PR(A)-NanoFIRE and PR(B)-NanoFIRE cell lines. This provided a system to investigate PR signalling within an ovarian cell context. Treatment of both PR(A)-NanoFIRE and PR(B)-NanoFIRE lines with dox and R5020 resulted in strong reporter induction relative to dox only treated cells at 9-fold and 39-fold, respectively (Figure 5B). Each provided a Z′ score greater than 0.7, confirming assay robustness and high signal separation upon PR activation with R5020. This demonstrated that aside from sensing HIF activity, NanoFIRE can be adapted to investigate the activity of tissue specific transcription factors in both endogenous and synthetic systems in a sensitive and signal-dependent manner.

## 4. Discussion

Here we describe a novel NanoLuciferase reporter system, NanoFIRE, which provides a robust and highly sensitive read out of HIF transcriptional activity in response to multiple stimuli and can be adapted to investigate other transcription factors and transcriptional regulatory pathways. We show that NanoFIRE is a robust and sensitive reporter system within a polyclonal setting across diverse cell lines and within primary cells, as demonstrated with granulosa cells of the ovary (Figure 2 and Figure 3). Across all cell lines and reporter constructs tested, NanoFIRE could be used to generate systems suitable for HTS. This was evident with all HIF and FIH systems providing a Z′ value greater than 0.5 in response to hypoxia mimetic treatment, demonstrating they are within the range of an excellent screen [20]. High Z′ values in response to hypoxia were also obtained for most cell lines tested, supporting the suitability of NanoFIRE for HTS also under hypoxia. 

Previous NanoLuciferase HIF reporter systems have either measured HIF-1 heterodimerisation through using HIF-1α and ARNT split NanoLuciferase fusions or expressed HIF-1α fused to full length NanoLuciferase [38,39]. In these systems, modulators of only either HIF-1 dimerisation or HIF-1α protein levels (i.e., PHD regulation), respectively, can be reported on. By HRE-NanoFIRE acting as a readout of HIF transcriptional activity, modulators of all aspects of control over the HIF pathway can be reported on, and the system can be adapted through changing the controlling response element to investigate other pathways, providing versatility which HIF fusion reporters lack. Additionally, the lentiviral nature of NanoFIRE provides versatility, particularly in difficult to transfect cells.

Compared to screening systems of the HIF pathway which transiently overexpress HIF fusion proteins in cells [38], use purified protein in vitro [32,40] or use molecular modelling based on known crystal structures [41], NanoFIRE has the advantage of being cell-based and also a direct readout of changes in endogenous HIF activity within a cellular context. This is particularly important, given there are multiple avenues of cell signalling, such as reactive oxygen species [42,43], lipopolysaccharide (LPS) [44], and numerous metabolites [45] that can influence HIF activity, all of which are commonly missed in overexpression or cell-independent screening systems. 

Given the high sensitivity and Z′ scores of the NanoFIRE system obtained across cell lines and signalling pathways, we envisage NanoFIRE as either a primary or secondary screening system to complement already-established fluorescent and transient firefly reporter systems in either small molecule screening or arrayed genetic screens for novel regulators of the HIF pathway. There remains a need for the discovery of novel HIF inhibitors, in particular for the treatment of cancer where the HIFs are commonly pro-tumorigenic by promoting angiogenesis and glycolysis to aid tumour growth and survival [5]. Unlike the discovery of HIF-2 inhibitors, which has been notably successful with the identification of PT-2385 for the treatment of renal cancers [41], there are a lack of direct HIF-1 inhibitors, with two of the better inhibitors, acriflavine and PX-478, showing non-specificity and indirect mechanisms of action [46,47,48]. In a small molecule screening setting, NanoFIRE complements established fluorescent-based HTS systems by providing a system not influenced by autofluroescent compounds and that overcomes the time delay required for fluorescent protein accumulation and maturation (Figure 2A,C) [49]. Furthermore, compared to firefly luciferase, NanoLuciferase has structural and substrate differences, thus allowing NanoFIRE to complement firefly-based systems for the screening and identification of common nuisance compounds, which can cause false positive and negative firefly readout [50,51]. The HEK-293T HRE-NanoFIRE line represents a sensitive reporter system specifically for HIF-1, given that HEK-293T cells do not express HIF-2α [9] (Figure 2A,B) and would thus be ideal for small molecule drug screening to identify novel compounds which specifically modulate HIF-1 activity, in particular those that directly inhibit HIF-1 which would have broad therapeutic potential in cancer.

Given the hypoxic insensitivity of the HRE-NanoFIRE system, it also has future applications in arrayed CRISPR genetic screening for identifying novel modulators of the HIF pathway under the most physiologically relevant conditions of hypoxia. This is advantageous over fluorescent-based systems which are typically used for genetic screening and either display reduced sensitivity or require an extensive reoxygenation period after hypoxic incubation [16,28]. Importantly, NanoFIRE may also have in vivo applications as NanoLuciferase is amenable to in vivo imaging [52,53,54], with recent research showing the development of alternative and more bioavailable substrates for sensitive NanoLuciferase imaging within deep tissues [53]. 

Adaption of NanoFIRE to investigate FIH and PR signalling (Figure 4 and Figure 5) demonstrated its use as a reporter system for various classes of transcriptional regulators, including synthetic transcription factors, and therefore as a tool for probing a wide range of signalling pathways. FIH-NanoFIRE provides a cell-based screening system for FIH regulators, as demonstrated with this line achieving a Z′ score greater than 0.9 in response to dox and DMOG. The poor response of FIH-NanoFIRE to the best available FIH-specific inhibitor DM-NOFD (Figure 4E) confirmed that alternative FIH inhibitors with better cell-based efficacy are required as a research tool, but may also have therapeutic potential, for example in the treatment of metabolic diseases [55,56] and of certain cancers [57,58]. The KGN PR(A)-NanoFIRE and PR(B)-NanoFIRE lines demonstrated that NanoFIRE can be used to dissect the transcriptional activity of exogenously expressed transcription factors (Figure 5) and is also sensitive enough to detect changes in activity of endogenously expressed PR. This highlights the broad versatility of the NanoFIRE reporter system, with further substitutions of the response element cassette offering a plethora of transcriptional pathways which could be investigated.

Finally, this system is highly adaptable, beyond the substitution of response elements driving expression of NanoLuciferase. While the system contains constitutively expressed EGFP which could be used for normalisation, as we have successfully done in our analogous dual fluorescent reporter system [13], this could also be interchanged for firefly or click beetle luciferase to form a dual luminescent reporter [21]. Additionally, the PEST-NanoLuciferase could be exchanged for secreted NanoLuciferase to allow assessment of temporal reporter expression in real time without cell lysis. Stable cell lines could also be used for the generation of monoclonal NanoFIRE lines, which we envisage would provide an even more consistent NanoFIRE reporter system with enhanced signal to background sensitivity.

In summary, NanoFIRE expands on the available repertoire of reporter systems and acts as a sensitive, highly versatile, stably integrating reporter for the analysis of HIF-dependent hypoxic signalling. NanoFIRE has future applications in the discovery of HIF modulators with therapeutic potential in human disease, and broad adaptability to investigate other signalling pathways.

## Figures and Tables

**Figure 1 biomolecules-13-01545-f001:**
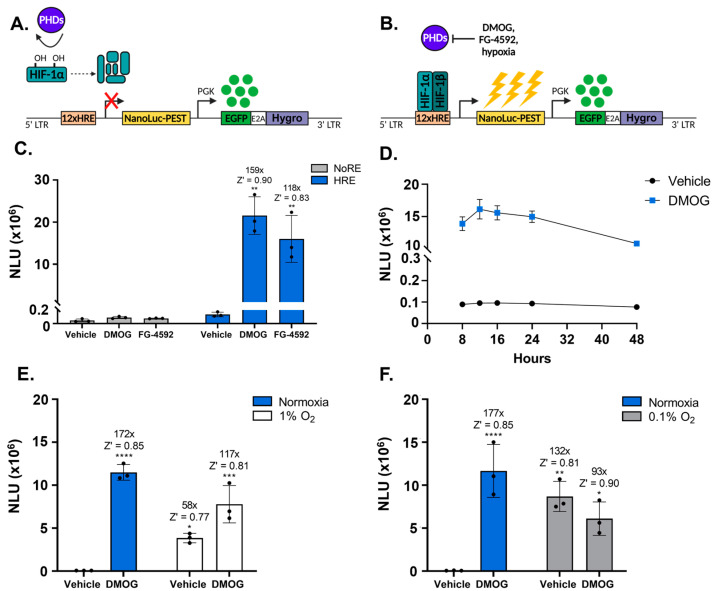
The U2OS HRE-NanoFIRE cell line provides a robust and sensitive readout of HIF transcriptional activity. Schematic of the HRE-NanoFIRE reporter construct and reporter activity under (**A**) conditions of normoxia and active PHDs, or (**B**) hypoxia or PHD inhibition with DMOG or FG-4592. NanoLuciferase Units (NLU) of U2OS HRE-NanoFIRE and U2OS NoRE-NanoFIRE cells treated for 16 h with (**C**) vehicle (0.2% DMSO), 1 mM DMOG or 50 μM FG-4592, (**E**) U2OS HRE-NanoFIRE cells under normoxia or 1% O_2_ with 1 mM DMOG or vehicle (0.1% DMSO), or (**F**) normoxia or 0.1% O_2_ with 1 mM DMOG or vehicle (0.1% DMSO). (**D**) NLU of U2OS HRE-NanoFIRE cells treated with vehicle (0.1% DMSO) or 1 mM DMOG for 8, 12, 16, 24, or 48 h. All cells seeded and assessed for reporter activity at the same time to ensure the same number of cells were assessed for each timepoint. *n* = 3 biologically independent experiments, each performed in triplicate, mean ± standard deviation, * *p* < 0.05, ** *p* < 0.005, *** *p* < 0.0005, **** *p* < 0.0001, one-way ANOVA with Tukey’s multiple comparisons. Statistics and fold change calculated relative to vehicle-treated cells under normoxia.

**Figure 2 biomolecules-13-01545-f002:**
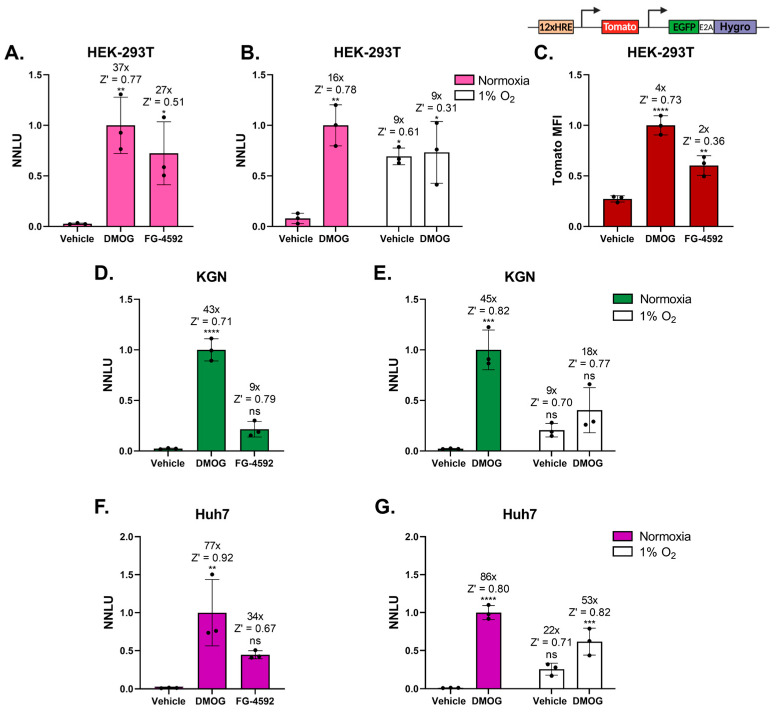
HRE-NanoFIRE can sense HIF reporter activity across numerous cell types and is more sensitive than fluorescent-based systems. Normalised NanoLuciferase Units (NNLU) of HEK-293T HRE-NanoFIRE cells treated for 16 h with (**A**) vehicle (0.2% DMSO), 1 mM DMOG, or 50 μM FG-4592 and (**B**) normoxia or 1% O_2_ with 1 mM DMOG or vehicle (0.1% DMSO). KGN HRE-NanoFIRE cells treated for 16 h with (**D**) vehicle (0.2% DMSO), 1 mM DMOG, or 50 μM FG-4592 and (**E**) normoxia or 1% O_2_ with 1 mM DMOG or vehicle (0.1% DMSO). Huh7 HRE-NanoFIRE cells treated for 16 h with (**F**) vehicle (0.1% DMSO), 0.1 mM DMOG, or 50 μM FG-4592 and (**G**) normoxia or 1% O_2_ with 0.1 mM DMOG or vehicle (0.1% DMSO). (**C**) Tomato Mean Fluorescent Intensity (MFI) of HEK-293T dual fluorescent reporter cells treated for 16 h with vehicle (0.1% DMSO), 1 mM DMOG, or 50 μM FG-4592. *n* = 3 biologically independent experiments, each performed in triplicate, mean ± standard deviation, * *p* < 0.05, ** *p* < 0.005, *** *p* < 0.0005, **** *p* < 0.0001, ns = not significant, one-way ANOVA with Tukey’s multiple comparisons. Statistics and fold change calculated relative to vehicle treated cells under normoxia. All NNLU and Tomato MFI values normalised to DMOG treated cells under normoxia.

**Figure 3 biomolecules-13-01545-f003:**
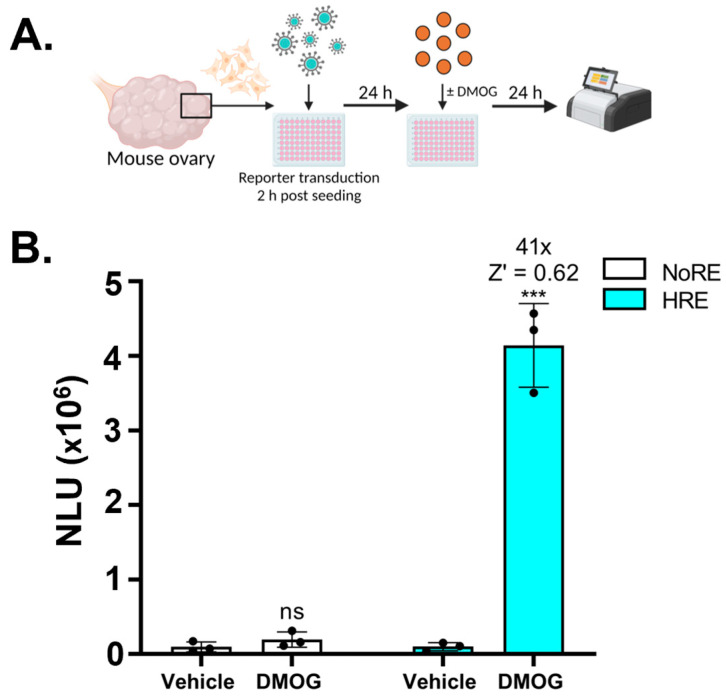
HRE-NanoFIRE can be used as a HIF reporter system in primary cells. (**A**) Workflow for mouse granulosa cell isolation, in vitro culture, and treatment prior to NanoLuciferase reporter assay. (**B**) NanoLuciferase Units (NLU) of primary mouse granulosa cells transduced with NoRE-NanoFIRE or HRE-NanoFIRE reporter virus and treated with vehicle (0.1% DMSO) or 1 mM DMOG for 24 h. *n* = 3 biologically independent experiments, each consisting of cells pooled from at least 3 mice and performed in duplicate or triplicate, presented as mean ± standard deviation. *** *p* < 0.0005, ns = not significant, *t* test assuming equal standard deviation. Fold change and statistics relative to vehicle.

**Figure 4 biomolecules-13-01545-f004:**
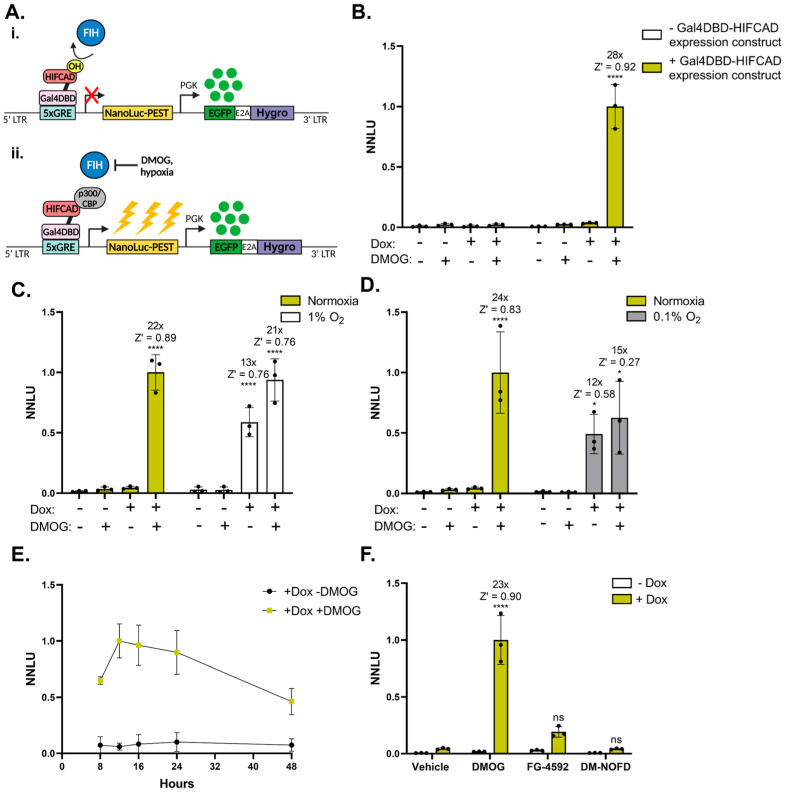
NanoFIRE can be adapted to investigate FIH-mediated transcriptional regulation. (**A**) Schematic of the FIH-NanoFIRE reporter system in doxycycline (dox)-treated cells under (**i**) conditions of normoxia and active FIH, or (**ii**) FIH inhibition with hypoxia or DMOG. (**B**) Normalised NanoLuciferase Units (NNLU) of HEK-293T GalRE-NanoFIRE cells (−Gal4DBD-HIFCAD expression construct) and FIH-NanoFIRE cells (+Gal4DBD-HIFCAD expression construct) treated for 16 h with 1 μg/mL dox or vehicle (0.1% H_2_O) and 1 mM DMOG or vehicle (0.1% DMSO). HEK-293T FIH-NanoFIRE cells treated for 16 h with (**C**) normoxia or 1% O_2_ with 1 μg/mL dox or vehicle (0.1% H_2_O) and 1 mM DMOG or vehicle (0.1% DMSO), (**D**) normoxia or 0.1% O_2_ with 1 μg/mL dox or vehicle (0.1% H_2_O) and 1 mM DMOG or vehicle (0.1% DMSO) and (**F**) 1 μg/mL dox or vehicle (0.1% H_2_O) and 1 mM DMOG, 50 μM FG-4592, 1 mM DM-NOFD, or vehicle (0.5% DMSO). (**E**) NNLU of HEK-239T FIH-NanoFIRE cells treated with 1 μg/mL dox and 1 mM DMOG or vehicle (0.1% DMSO) for 8, 12, 16, 24, or 48 h. All cells seeded and assessed for reporter activity at the same time to ensure the same number of cells were assessed for each timepoint. *n* = 3 biologically independent experiments, each performed in triplicate, mean ± standard deviation, * *p* < 0.05, **** *p* < 0.0001, ns = not significant, one-way ANOVA with Tukey’s multiple comparisons. Statistics and fold change calculated relative to dox only treated cells under normoxia. All NNLU values normalised to dox- and DMOG-treated cells under normoxia.

**Figure 5 biomolecules-13-01545-f005:**
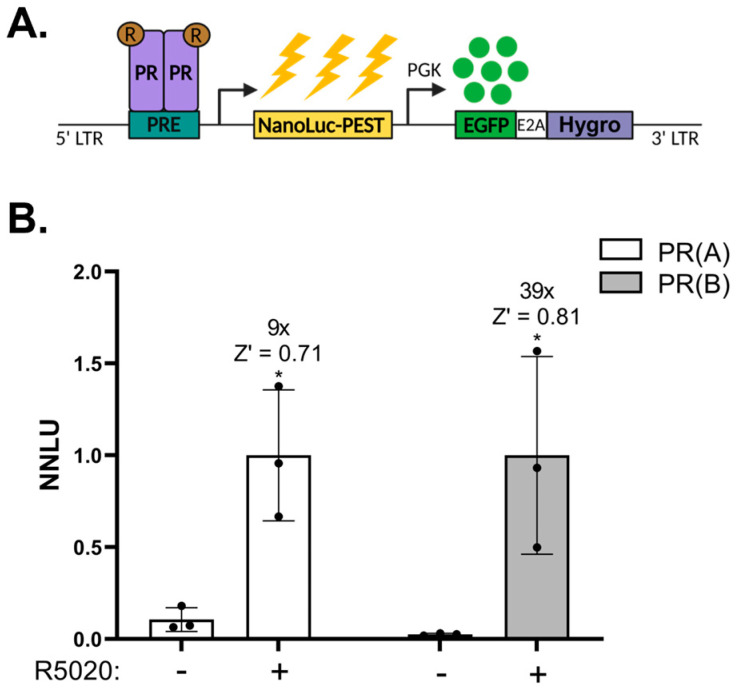
NanoFIRE can be adapted to investigate Progesterone Receptor (PR) transcriptional activity. (**A**) Schematic of the PRE-NanoFIRE reporter construct and reporter activity under conditions of R5020 (R) and doxycycline treatment for PR expression. (**B**) Normalised NanoLuciferase Units (NNLU) of KGN PR(A)-NanoFIRE and KGN PR(B)-NanoFIRE cells treated for 16 h with 100 nM R5020 or vehicle (0.1% ethanol) and 1 μg/mL dox. Values normalised to R5020 treated cells for each line. *n* = 3 biologically independent experiments, each performed in duplicate, mean ± standard deviation, * *p* < 0.05, unpaired *t* test for each individual cell line. Fold change relative to −R5020 (dox only) treated cells stated.

## Data Availability

The data presented in this study is openly available in FigShare at https://doi.org/10.25909/24323299.v1.

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
