# Peer review of "NanoFIRE: A NanoLuciferase and Fluorescent Integrated Reporter Element for Robust and Sensitive Investigation of HIF and Other Signalling Pathways"

_biomolecules, 2023, doi:10.3390/biom13101545_

Round 1

Reviewer 1 Report

This is a very interesting study by Roennfeldt and co-authors. The experiments are well-designed, all necessary controls are included, and the presentation is concise. 

Potentially, the authors could comment on the analysis of PHD specific inibitors with Nano-FIRE since FIH is somewhat promiscuous and, therefore, rarely regarded as a target of pharmacological inhibition. 

Author Response

We thank the reviewer for their positive comments and agree that PHD-specific regulators have more obvious and important pharmacological application than FIH-specific regulators. To address this, we have included additional text highlighting the importance of PHD/HIF-1 specific regulators and toned down the discussion of pharmacological inhibitors of FIH (see section 4. Discussion, page 13, lines 529 and 547, highlighted).

Note that while FIH may be promiscuous with numerous ankyrin repeat and other target proteins, it has been shown to play important roles in metabolic regulation and cancer progression and is therefore a potential therapeutic target.

Reviewer 2 Report

Alison Roennfeldt et al present and characterize a new, highly sensitive, and versatile reporter system, NanoFIRE, a NanoLuciferase and Fluorescent Integrated Reporter Element that, under the Hypoxic Response Element, is a sensor of HIF activity in response to both chemical inducers of the HIF pathway and hypoxia.

The research project is well designed, the experimental procedures are clear enough and the results are adequately presented and discussed.

The manuscript deserves publication in Biomolecules after the following minor points are discussed and revised by the authors:

-        It is not clear to me with respect to which sample the statistical significance is indicated. I suppose the authors compared all samples to the vehicle-treated cells under normoxia. In this case, it could also be useful to compare the results in the same experimental condition;

-        Is there a reason why they didn’t perform experiments in hypoxia with primary cells?

Author Response

We thank the reviewer very much for the positive comments and constructive feedback regarding the manuscript. 

Specific Points:

1. The reviewer is correct, and the statistical tests were performed on samples compared to the vehicle-treated cells under normoxia. We apologise for any confusion and have therefore included additional information in each of the relevant figure legends for clarification.

Note that it was induction in response to treatment that was of primary interest and hence analysed statistically. So, we have intentionally not included additional statistical tests of samples under the same experimental conditions as this would unnecessarily complicate the figure and make it more difficult for a reader to understand and interpret.

2. Primary cells were very difficult to obtain and in very limited supply due to animal and ethics limitations. So, we only performed the critical experiments with DMOG to show that the system responds robustly in primary cells, and did not have cells available for any additional, less important experiments.